# Cell-Type-Specific Quantification of a Scaffold-Based 3D Liver Co-Culture

**DOI:** 10.3390/mps3010001

**Published:** 2019-12-23

**Authors:** Marc Ruoß, Vanessa Kieber, Silas Rebholz, Caren Linnemann, Helen Rinderknecht, Victor Häussling, Marina Häcker, Leon H. H. Olde Damink, Sabrina Ehnert, Andreas K. Nussler

**Affiliations:** 1Department of Traumatology, Siegfried Weller Institute, BG-Klinik Tübingen, Eberhard Karls University, 72076 Tübingen, Germany; vanessa@kieber.de (V.K.); silasrebholz@gmail.com (S.R.); caren.linnemann@student.uni-tuebingen.de (C.L.); helen.rinderknecht@student.uni-tuebingen.de (H.R.); victor.haeussling@student.uni-tuebingen.de (V.H.); marina.haecker@web.de (M.H.); sabrina.ehnert@gmail.com (S.E.); andreas.nuessler@gmail.com (A.K.N.); 2Matricel GmbH, 52134 Herzogenrath, Germany; olde_damink@matricel.de

**Keywords:** quantification, 3D culture, co-culture, cell number, PCR-based method

## Abstract

In order to increase the metabolic activity of human hepatocytes and liver cancer cell lines, many approaches have been reported in recent years. The metabolic activity could be increased mainly by cultivating the cells in 3D systems or co-cultures (with other cell lines). However, if the system becomes more complex, it gets more difficult to quantify the number of cells (e.g., on a 3D matrix). Until now, it has been impossible to quantify different cell types individually in 3D co-culture systems. Therefore, we developed a PCR-based method that allows the quantification of HepG2 cells and 3T3-J2 cells separately in a 3D scaffold culture. Moreover, our results show that this method allows better comparability between 2D and 3D cultures in comparison to the often-used approaches based on metabolic activity measurements, such as the conversion of resazurin.

## 1. Introduction

In recent years, both the cultivation of liver cells in 3D cultures and their cultivation in co-culture with other cell types have been described in various studies [1,2]. These approaches aim to maintain the metabolic activity of primary hepatocytes or to increase the metabolic activity of liver model hepatoblastoma cell lines, such as HepG2 cells [3,4,5,6]. However, as the cultivation systems become more complex, difficulties arise. In conventional 2D culture, the quantification of cells can be done relatively quick in various ways. In addition to counting the cells by the classical trypan blue exclusion method, quantification of cells can be done by measuring the activity of mitochondrial dehydrogenases using substrates (e.g., resazurin, MTT, or XTT), ATP levels, total protein content (e.g., by sulforhodamine B (SRB) staining or Lowry measurement), or the DNA content (e.g., by using DNA staining dyes like CyQuant or Hoechst 33342) [7,8,9,10,11,12,13,14].

All these methods cited have their advantages and disadvantages (described in Table 1) [15]. The main disadvantage of most methods is that their use in 3D culture is limited or not possible at all. Furthermore, cell lysis is needed in all methods, except for the measurement of resazurin turnover. Additionally, interactions of the test reagent with scaffold ingredients could affect the results of the quantification between 2D and 3D and such an interference cannot easily be identifie.

Measurement of the mitochondrial activity (e.g., by measurement of resazurin conversion) is often used for quantification of 3D approaches, but it has its limitations since the results of 2D and 3D approaches are often not comparable [20]. The activity of mitochondrial dehydrogenase in these assays (resazurin, MTT, or XTT) gives only an estimation of the combination of metabolic activity and redox potential of the cultured cells [21]. In order to use resazurin conversion for quantification, the stress level and redox potential of the cells between different conditions have to be constant. When comparing 2D to 3D culture, surface properties are one factor that can modify the stress level of the cells [22]. Therefore, constant conditions cannot be assumed. The same is true for matrix stiffness. Scaffolds often mimic natural organ stiffness (e.g., in the liver ≤6 kPa [23]), while standard tissue culture plastic with ~100 MPa is much stiffer [24].

Thus, it is necessary to develop a normalization method that is mostly independent of external factors, such as cell stress, pH, or stiffness. Only then a realistic comparison between 2D and 3D culture is possible. Since the DNA content in all cells is the same independent of the cultivation condition, we decided to use DNA to quantify and compare our 2D and 3D results. In this study, we tested different approaches of DNA quantification of 2D and 3D cultures and compared the results to the commonly used measurement of resazurin conversion. Additionally, we developed a PCR-based measurement of the DNA content as a normalization method of our 2D and 3D co-culture assays. We used murine 3T3-J2 cells in co-culture with HepG2 cells to increase the liver cell metabolic activity. The 3T3-J2 cells are an established model to increase not only the metabolic capacity of HepG2 cells but also of hepatocytes [25,26,27]. By using species-specific primers, we succeeded to quantify the individual numbers of the different cell types. This approach can be useful to study possible cell–cell and cell–matrix interactions of different cell types in 3D co-cultures. The aim of the study and the quantification approaches tested are summarized in Figure 1.

## 2. Materials and Methods

### 2.1. Cell Culture and Cell Seeding

The cultivation of both cell types used was carried out in an incubator at 37 °C and 5% CO_2_ in a humidified atmosphere. For the cultivation of HepG2 cells, DMEM with high glucose (4.5 g/L) medium (Sigma-Aldrich, Munich, Germany) containing 10% fetal calf serum (FCS) (Thermo Fisher Scientific, Waltham, MA, USA) and 1% penicillin-streptomycin (P/S) (10,000 units penicillin and 10 mg streptomycin/mL) (Sigma-Aldrich) was used [28]. For the culture of 3T3-J2 cells, DMEM medium containing 10% bovine calf serum (BCS) (Sigma-Aldrich) and 1% P/S was used [26]. For the experiments, cells cultivated in cell culture flasks were washed once with PBS followed by incubation for 10 min at 37 °C with trypsin/EDTA solution (0.5 g/L trypsin and 0.2 g/L EDTA). The detachment of the cells was checked microscopically. When cell detachment was complete, DMEM medium containing 10% FCS was added to stop the reaction. The cells were transferred in a reaction tube and were centrifuged at 600× *g* for 10 min. The supernatant was removed, and the cells were resuspended in a defined amount of DMEM medium containing 10% FCS and 1% penicillin-streptomycin. The cell number was determined using a Neubauer chamber. For the comparison of different quantification techniques, we used HepG2 and 3T3-J2 cells in mono-culture. HepG2/3T3-J2 cells (1, 0.5, 0.25, and 0.125 × 10^5^) were plated in 24-well plates for the comparison of the different quantification techniques. For testing our newly developed co-culture quantification approach, we used constant cell numbers of 0.5 × 10^5^ cells for mono-culture. In the co-cultures, 0.5 × 10^5^ cells for each cell type were used. All experiments in 2D and 3D culture were carried out in 24-well plates using high glucose DMEM medium (containing 10% FCS and 1% P/S).

For 3D culture, Optimaix-3D scaffolds (Matricel, Herzogenrath, Germany) and self-made cryogels were used. For optimal cell attachment on the Optimaix-3D scaffold, the so-called “drop-on” seeding method was used [4]. Therefore, the cell suspension was concentrated by centrifugation to obtain a cell density of 3.33 × 10^6^ cells/mL. For both cell types, serial dilutions were prepared. For mono-culture, 30 µL of the respective cell solution was added on top of each scaffold (prepared in a well of a 24-well plate). For co-culture, 30 µL of a cell solution containing both cell types were added on top of the scaffolds. After an attachment period of 4 h, additional medium was added to obtain a total volume of 500 µL in all conditions. For our self-made cryogels, we increased the volume (but not the cell number) of the cell solution, since this scaffold was larger (10 mm in diameter). The volume of the cell solution was increased to 40 µL to achieve a uniform distribution. Furthermore, the total volume of the medium was adapted to 700 µL.

### 2.2. Cell Quantification by Optical Methods

The quantification of cell numbers under the different conditions was carried out 18 h after seeding. For our self-made scaffold, we reduced this period in the course of the study to 12 h to avoid possible influence due to different doubling times of the cells caused by the culture conditions. For cell quantification, resazurin conversion and DNA content (absorption- and fluorescence-based with Hoechst 33342 and CyQuant) were measured. In addition, quantification of the species-specific DNA content was tested by PCR-based methods.

#### 2.2.1. Resazurin Conversion

As previously described, measurement of mitochondrial dehydrogenase activity is often used to quantify cells. Resazurin is particularly suitable for the 3D culture since the water-soluble product is released into the supernatant. To measure resazurin conversion, the scaffolds were transferred into a new 24-well plate to avoid the influence of cells attached to the plate surface. The medium of the 2D cultures was also removed. A 0.0025% resazurin solution in medium was added and, after incubation for 1 h at 37 °C, the formed resorufin was quantified (fluorescence) at a wavelength of 544 nm/590–10 nm using the OMEGA Plate Reader (BMG Labtech, Ortenberg, Germany) [4].

#### 2.2.2. DNA Isolation in 2D and 3D Scaffold Cultures

Previous experiments have proven that it is impossible to collect all living cells from the scaffold. Treatment with trypsin is unsuccessful because FCS from remaining medium (even after washing) inactivates the enzyme. Therefore, we decided to isolate the DNA directly from the scaffolds, using a modified protocol developed initially for DNA extraction from tissue [29]. For extraction of DNA from cells plated on scaffolds, the scaffolds were first washed with PBS. Two scaffolds of each group were pooled for further DNA isolation. To remove disturbing fluid from the scaffolds, they were transferred to a cell strainer and centrifuged at 600× *g* for 10 min before being transferred to a 2 mL reaction tube. Supernatants were discarded. Detached cells, which can be found after centrifugation as a pellet in the reaction tube, were resuspended in 250 µL 50 mM NaOH solution, which was then added to the scaffolds in a new reaction tube. The cells in 2D culture were also washed with PBS and then detached from the plate by using the same amount of heated (98 °C) 50 mM NaOH solution for 5 min. Cell detachment was verified by microscopy. For DNA extraction, the cells/scaffolds were incubated at 98 °C for 30 min. Subsequently, the reaction tubes were vortexed thoroughly and briefly frozen at –80 °C. Vortexing and freezing improved cell lysis. To all thawed samples, 250 μL ddH_2_O and 25 μL Tris/HCl (1 M, pH = 8) were added. These DNA lysates were used for all tested DNA-based quantification methods. Since our self-made scaffold had a higher water uptake, we doubled the volumes of NaOH solution, ddH_2_O, and Tris/HCl that were used.

#### 2.2.3. Absorption Measurement by Using LVIS Micro Drop Plate

For the absorption-based quantification, all samples were measured on the LVIS Plate (BMG Labtech). One run consists of two steps. For the first part, a blank was measured. In our case, it was 2 µL of DNA isolation buffer (1mL demineralized water, 1 mL NaOH (50 mM), and 100 µL Tris/HCl (1 M, pH = 8)). In the second step, 2 µL of each sample was measured in duplicate. The DNA concentrations were calculated by the BMG Labtech OMEGA (Ortenberg, Germany) analyzation software MARS.

#### 2.2.4. Fluorescence-Based CyQuant Measurement

For the fluorescence-based CyQuant measurement, we adapted the manufacturer’s protocol. Briefly, the CyQUANT™ Cell Proliferation Assay Dye (Thermo Fisher Scientific, Waltham, USA) was diluted 1:400 in our DNA isolation buffer, which consisted of 250 µL demineralized water, 250 µL NaOH (50 mM), and 25 µL Tris/HCl (1 M, pH = 8). The detection range was verified with the DNA standard from the CyQuant kit. A total of 10 µL of each sample was pipetted into a 96-well plate in duplicate. Afterwards, 100 µL of staining solution (CyQuant in DNA isolation buffer) was added to each well. The fluorescence measurements were carried out at an excitation wavelength of 485–12 nm and an emission wavelength of 520 nm.

#### 2.2.5. Fluorescence-Based Hoechst 33342 Measurement

For the Hoechst 33342 measurement, we used a modified protocol following Richards et al. [30]. Therefore, Hoechst 33342 was diluted in a stock concentration of 2 mg/mL in PBS. This stock solution was diluted 1:100 in the earlier described DNA isolation buffer. A total of 90 µL of each sample was pipetted into a 96-well plate in duplicate before 10 µL staining solution was added per well. The samples were measured (fluorescence) at an excitation wavelength of 355 nm and an emission wavelength of 460 nm.

### 2.3. Cell-Type-Specific DNA Quantification

#### 2.3.1. Test of Different Primers for the Usability in a Species-Specific DNA Quantification Method

For species-specific DNA quantification, it is necessary that the used primers are completely species-specific. An amplicon length of 150 to 250 bp is beneficial because it enables use in both conventional PCR and qPCR. Additionally, the used primers are not allowed to be exon spanning. Our criteria-fulfilling primers can be found in Table 2. By using primer blast (National Center for Biotechnology Information, U.S. National Library of Medicine USA), the species-specificity of the sequences was ensured. To avoid products on potentially unintended templates in the other species, we verified the species-specificity experimentally (Appendix A). Additionally, we used the highest cell number of the mono-culture of the respective other cell line as a negative control for the quantification of our co-culture experiments.

#### 2.3.2. Conventional Semi-Quantitative PCR

PCR reactions were carried out by using the Red HS Taq Master Mix from Biozym (Vienna, Austria) according to the manufacturer’s instruction. Briefly, a master mix was prepared that contained 10 µL Biozym Red HS Taq Master Mix, 1 µL each of forward primer and reverse primer (final concentration 400 nM), and 4 µL DEPC water for a single 20 µL PCR reaction. After distribution of the master mix to PCR tubes (16 µL each), 4 µL template DNA was added. The PCR was performed with the following program: initial denaturation 2 min at 95 °C; for the amplification, 30 cycles of the following steps: 15 s denaturation at 95 °C, 15 s annealing (temperature is primer-dependent (see Table 2)), and 45 s extensions at 72 °C, final denaturation 10 min at 72 °C. For analysis of the results, 8 µL of each sample was loaded onto a 2% agarose gel, which contained ethidium bromide for DNA staining. For separation of the samples, gel electrophoresis was carried out (80 V for 45 min). As referenced for DNA molecular weights, the DNA-Marker pUC19/Msp I (Carl Roth, Karlsruhe, Germany) or the Bioline Hyperladder II (Bioline, Memphis, TN, USA) were used. Intensity of the bands was measured with ImageJ software version 1.5 (National Institutes of Health, Bethesda, MD, USA). To analyze the PCR products in the logarithmic phase of the amplification, the cycle number and amount of template were optimized (Appendix A).

#### 2.3.3. Quantitative Real-Time PCR

In addition to conventional PCR, quantitative real-time PCR was performed. Therefore, Step One Plus^®^ Real-Time PCR System (Life Technologies, Carlsbad, CA, USA) and GreenMasterMix, High ROX (Genaxxon Bioscience, Ulm, Germany) were used [19]. The same primer and DNA concentrations used for conventional PCR were used here. qPCR was carried out with the following parameters: 15 min at 95 °C for initial denaturation; 40 cycles of amplification with the following steps: denaturation for 15 s at 95 °C, annealing for 30 s at 62 °C, and extension for 30 s at 72 °C. Evaluation of the results was performed using StepOne Software version 2.3 (Life Technologies).

### 2.4. Cell-Type-Specific Cell Labeling

In addition to resazurin conversion measurement and the isolation of DNA, we took light microscopy pictures of the cells using the cell concentrations previously described. For better visualization of the cells in 3D culture and distinction between the two cell types in co-culture, the 3T3-J2 cells were stained using red fluorescent Cytoplasmic Membrane Staining Kit (PromoCell, Heidelberg, Germany), according to the manufacturer’s instructions. Briefly, 3T3-J2 cells were detached by treatment with trypsin/EDTA and diluted to a concentration of 1 × 10^6^ 3T3-J2 cells. A total of 10 µL of cell labeling solution was added per mL of cell suspension. Cells were then incubated for 10 min at 37 °C and afterwards centrifuged at 600× *g* for 10 min. The supernatant was removed, the cells were washed once with PBS, and then resuspended in pre-warmed medium. Stained 3T3-J2 cells (0.5 × 10^5)^ and 1 × 10^5^ unstained HepG2 cells were seeded in 2D and on the Optimaix-3D scaffold as described before. After approximately 18 h, the nuclei were stained with Hoechst 33342 (1:1000 dilution, 2 µg/mL final concentration). Pictures were taken by fluorescence microscopy (EVOS FL, Life Technologies, Darmstadt, Germany).

#### Statistics

Statistical significance between two different groups was evaluated by the non-parametric Mann–Whitney U test. Statistical significance of more than two groups was evaluated by the non-parametric Kruskal–Wallis H test followed by Dunn’s multiple comparison test (GraphPad Prism 8.00 Software, San Diego, CA, USA). All data are presented as mean ± SEM of at least three independent experiments (*n* ≥ 3). All statistical comparisons were performed two-sided using *p* < 0.05 (*), *p* < 0.01 (**), and *p* < 0.001 (***) as levels of significance.

## 3. Results

This study aimed to develop a simple and precise method for the quantification of a scaffold-based co-culture system. For improving the metabolic activity of human hepatic cells (e.g., HepG2 but also primary hepatocytes), a co-culture with fibroblasts (e.g., murine 3T3-J2 cells) was carried out. Planning to continue the work with a combination of these cells, we selected a combination of HepG2 cells and 3T3-J3 cells as a test model. The design of the study will allow a rapid transfer into other co-culture systems.

### 3.1. Morphological Differences of HepG2 and 3T3-J2 Cells Can Be Used with Restrictions for Quantification in 2D Co-Culture but Not in 3D Co-Culture

As shown in Figure 2, in principle, the morphological differences of the two cell types in 2D culture can be used for a cell-type-specific quantification. While the fibroblasts have a spindle-shaped morphology, HepG2 cells are more hexagonal. The limitation of this eye-based quantification is quickly reached when cells grow dense or in 3D culture. Furthermore, the microscopic counting of the cells is very time-consuming. When adding fluorescent dyes into this quantification method, visualization and distinction of both cell types become easier, and quantification can be automated using software tools [31,32,33]. But this method is still not applicable to 3D scaffold cultures since cells inside the scaffold cannot be counted. Figure 2 shows the microscopic pictures and fluorescent staining of mono- and co-cultures in 2D and 3D.

### 3.2. Quantification of Conventional 2D Culture and 3D Scaffold Culture by Measuring the Mitochondrial Activity

Measurement of resazurin conversion, or similar methods such as MTT or XTT, has been widely used in the past to quantify cells in 3D scaffold culture [4,6]. As shown in Figure 3 (in detail in Appendix A, for each cell type separately including error bars), a correlation between increasing resazurin turnover and increasing cell counts was found in both systems. Therefore, the conversion of resazurin could be applied to our 2D system as well as in the 3D system. Regardless of the cell type, a significant difference in the resazurin conversion can be observed between the 2D culture and the corresponding 3D culture. To compare the different quantification methods, we calculated the area under the curve (AUC) of the 2D and 3D cultures, which allowed a method-independent comparison of the different quantification techniques.

### 3.3. Comparison of Alternatives to Resazurin Conversion for the Quantification of Conventional 2D Culture and 3D Scaffold Culture

In order to investigate whether the observed differences in resazurin conversion are due to different metabolic activities or different adherence of the cells in 2D and 3D cultures, we looked for a quantification approach that is mostly independent of external factors (e.g., cell stress). Since each cell contains the same amount of DNA and DNA is physically stable, we decided to test the quantification of cell numbers via quantification of the DNA.

#### 3.3.1. Comparison of Different Approaches for Cell Detachment in 2D and from the Optimaix-3D Scaffold

In order to quantify the amount of DNA, the cells or the DNA have to be effectively retrieved from the scaffold. To determine the best approach for cell detachment in 3D, 1 × 10^5^ HepG2 cells were plated in 2D and 3D. On the next day, the scaffolds were washed with PBS once. In the following, the cells or their DNA were removed from the scaffolds using one of three different approaches: centrifugation for 10 min at 600× *g*, and incubated with trypsin/EDTA for 10 min or treatment with NaOH as described in 2.2.2. As shown in Figure 4, effective removal of DNA is only possible by using NaOH. Only there an amount of DNA comparable to 2D can be obtained. Therefore, this method was used for further experiments.

#### 3.3.2. Comparison of Methods for the Quantification of Conventional 2D Culture and 3D Scaffold Culture

Different DNA quantification methods were tested including an absorption-based measurement of DNA concentrations and two fluorescence-based methods (Hoechst 33342 and CyQuant). The tested fluorescent dyes showed a strong increase in fluorescence when binding to DNA. Figure 5 (in detail in Appendix A, separated for each cell type with error bars) shows that the results of all tested DNA-based methods differ significantly from the results of the resazurin measurements. While the conversion of resazurin in 3D cultures was less than half that of 2D cultures, the measured amount of DNA in 3D cultures was comparable or higher than that of 2D cultures, regardless of the method.

Comparing the single methods, we observed for the absorption-based measurement that the 3D cultures showed higher absorption values than the 2D cultures (Figure 5a). The measurement using CyQuant showed a contrary result (Figure 5c). Measurement with Hoechst showed nearly the same amount of DNA in both 2D and 3D cultures (Figure 5b). Except for the absorption-based measurements in 3D cultures, all tested methods showed a correlation between the increase in cell number and the increase in the measured signal. These results proved that isolation of DNA from 3D scaffold cultures and 2D cultures were feasible with the DNA isolation method used.

However, all these methods do not allow cell-type-specific quantification, since only the total DNA content is measured. Therefore, a PCR-based method for quantification was tested. By using human and mouse cell lines in the test system, a species-specific signal can be generated by using species-specific DNA primers. The complete specificity of the used primers (*hUGT1A6* and *mIL-11*) used was checked with primer blast and by preliminary experiments (Appendix A). Figure 5d clearly shows a correlation between cell number and the measured cycle threshold (Ct) value when using the qPCR-based quantification method. Independent of the cell type, no difference between 2D and 3D cultured cells could be detected. Since this method can also be used for cell-type quantification, we selected it for the evaluation of the following co-culture experiments.

For better comparison of the used methods, three independent standards of both cell lines (HepG2 and 3T3-J2) (Appendix A) in the range of 1, 0.5, 0.25, 0.125, and 0.0625 × 10^5^ cells were prepared. The following standards were used to create standard curves for each method separately (three independent standard curves at least in duplicates). The standard error of the y-intercept and the slope of the linear regression were used to calculate the analytical figures of merit. As shown in Table 3, all methods showed a Limit of Quantitation (LOQ) > 2 × 10^3^ cells regardless of the cell type. The fluorescence-based CyQuant measurement and the qPCR-based method show the lowest Limits of Detection (LOD) and LOQ values. Since only the qPCR-based approach allowed the discrimination between the two tested cell lines, this method was used in the further course of the study.

### 3.4. PCR-Based Co-Culture Quantification

For co-culture experiments, we used the same number of cells for all conditions (co-culture and mono-culture of both cell types, each in 2D and 3D). To avoid saturation of the scaffold or the plate with cells, we used 5 × 10^4^ 3T3-J2 cells and 5 × 10^4^ HepG2 cells at each culture condition. We measured the conversion of resazurin as an established method for co-culture quantification. In addition, we also carried out qPCR and conventional semi-quantitative PCR of DNA samples (for quantification of each cell type independently). As shown in Figure 6a, resazurin conversion of cells in co-culture exceeds that of the combination of the respective mono-cultures, regardless of whether the cells were plated in 2D or 3D. This result was initially surprising. Due to the limited space for each cell available in co-culture, we assumed that herein total fewer cells would adhere to the scaffold/plate. However, we found the opposite in the measured resazurin conversion (Figure 6a). This result is also accompanied by a higher amount of measured DNA, which could be observed especially in the 3T3-J2 cells in the co-culture approach (Figure 6c,d, right). As Figure 6b clearly shows, a cell-type-specific quantification was possible with the selected primers, since the corresponding signal in the controls (5 × 10^5^ cells of the other cell type, lane C) is missing. Overall, the qPCR and semi-quantitative PCR showed nearly the same results. The qPCR displayed a higher scattering, which resulted in a higher standard deviation.

Next, we verified whether or not this quantification method could be translated to other scaffolds. Therefore, we tested the quantification method on our recently developed HEMA-based cryogel scaffold. Since this scaffold has a larger diameter and a higher volume (see Material and Methods), an adaptation of the volumes for the DNA isolation was necessary. To reduce differences between the culture conditions, which are possibly caused by different proliferation rates under the varying conditions, we reduced the cultivation time for this experiment from 18 h to 12 h.

As the results in Figure 7 show, the conversion of resazurin in the HEMA-based 3D cryogel was independent of the cell line significantly lower than in the 2D cultures. Regardless of the cell type, no significant differences were found in the PCR-based method between the 2D and 3D cultures. The amount of cell-type-specific PCR products in the 2D co-cultures was also higher than in the respective mono-cultures (Figure 7b,c, right), but this effect was significantly lower compared to the results shown in Figure 6. In general, a clear correlation between the number of cells and the measured PCR signals could be demonstrated. Species-specificity of the primers could also be confirmed, which proved the cell-type-specificity of this method (Figure 7b,c). Overall, the quantification method could be transferred to the self-made scaffolds by modification of the amounts of DNA isolation buffer.

## 4. Discussions

The cultivation of cells in 3D systems or of different cell types in co-culture systems are two approaches that often better mimic the in vivo environment [1,2]. Both approaches can also be combined [34], resulting in a system where both cell–cell and cell–matrix interactions can be modeled [35,36]. However, especially 3D systems are associated with limitations in the analysis methods [37]. In terms of normalization, many quantification methods that are successfully used in conventional 2D cultures cannot be used or are of limited use in 3D cultures [7,12,18]. A detachment of the cells from a 3D matrix, which would simplify the analysis, is often not possible without destroying the cells [2]. Therefore, normalization of mitochondrial dehydrogenase activity (resazurin, MTT, or XTT) or measurements of intracellular ATP levels are currently commonly used to quantify 3D culture experiments [4,6,8]. But these quantification methods do not allow direct comparison between 2D and 3D cultures, which is important for a comparison of the results of both cultivation techniques [38]. Metabolic activity assays, such as resazurin conversion, are highly susceptible to cell stress, which can easily be affected by culture conditions, such as oxygenation, temperature, 3D scaffolds, etc. [39]. The present work shows, when applying the fluorescent dye Hoechst 33342, it is easily possible to quantify various cell numbers in 2D and 3D mono-cultures. The measurement of the DNA amount with an absorption measurement or the alternative fluorescence-based CyQuant measurement differs from the results determined by Hoechst 33342, PCR, and qPCR. Absorption-based DNA measurement is sensitive to scaffold components released during culture or washing steps. This might explain the differences seen between the DNA content in our 2D and 3D cultures when quantified photometrically. In the case of the CyQuant measurement, it is known that FCS and phenol red from residual medium can lower the measured values, giving an explanation for the decreased DNA amounts measured in 3D cultures. Optimization of the CyQuant protocol could improve its suitability for use with 3D cultures. However, the major drawback of this method is the absence of cell-type-specific quantification in a co-culture system. A quantification just based on the plated cell number is not possible since it is not predictable whether the cells behave differently in 2D and 3D mono- or co-cultures [40,41]. Therefore, prior to establishing a scaffold-based co-culture model, it seems essential to develop a method that allows cell-specific quantification and thus allows an interpretation of functional results. To our surprise, no suitable method was available in the literature, except using flow cytometry with cell-type-specific surface markers and FISH analysis with specific gene sequences [42,43,44,45,46]. But both of these techniques require living/intact cells in suspension, which is impossible in scaffold cultures [2].

Another method that can be used for quantification of a co-culture is the (computer-based) analysis of microscopic images, which has been successfully established for 2D cultures [31,32,33] but cannot be used in 3D scaffold cultures since the scaffold-penetrating cells cannot be detected.

Thus, we searched in the literature for an alternative approach. Several studies using pellet cultures for the differentiation of Mesenchymal stem cells (MSCs) into cartilage cells determined the ratio of MSCs from different species by qPCR [41,47]. Another work has shown that it is possible to determine the ratio of different bacteria strains in a biofilm by species-specific primers [48]. Based on the latter, we adapted this technology to our needs. As shown in our results, the species-specific quantification with the adapted PCR-based method is possible. The method is very sensitive and can detect signals for less than 10,000 cells. For cell-specific quantification of a co-culture of human HepG2 cells with murine 3T3-J2 cells, PCR-based methods were the best choice. Semi-quantitative PCR and qPCR allowed for species-specific quantification. The transfer to another scaffold was also possible, underlining the universality of this approach.

Interestingly, when using 3T3-J2 cells in co-culture over 18 h, a higher amount of DNA compared to the respective mono-culture could be measured. Based on our data, we cannot state whether this was due to a promotion of cell division by the co-culture or due to improved adherence in the co-culture. The effect is less strong after 12 h, suggesting a proliferative effect by the co-culture system alone. Paracrine or autocrine actions that stimulate cell proliferation could be responsible for this effect [49]. A positive effect of 3D culture and co-culture on cell attachment or proliferation was also described by Fasolino et al. [50]. These results clearly underline the importance of cell-type-specific quantification of different cell types in a co-culture system in order to validate the effects of the same cell type generated in conventional mono-culture.

One limitation of this system is the dependency on different gene sequences. For example, it is not yet possible to quantify a co-culture model with primary human hepatocytes and non-parenchymal cells from the same donor. However, Callaghan et al. suggested to overcome this problem by determining the telomere length of different somatic cell types [51,52]. DNA barcoding is another interesting technique that enables tracking of cells from different origins in co-cultures. Therefore, the cells are transduced with a library of viral vectors. This theoretically allows the tracking of clones from each individual cell in the co-culture approach [53,54,55]. For data evaluation, DNA sequencing of cells is necessary, which makes this interesting approach quite expensive and not feasible for many laboratories. Another alternative might be the stable integration of foreign genes into cells of interest, for which specific primers could then be used. One example of this approach is immortalized hepatocytes [56]. For immortalization, primary human hepatocytes are transduced with virus particles containing the DNA sequence of human papillomavirus (HPV16) E6/E7 [56]. Alternatively, cells can be transfected with fluorescence proteins like GFP or mCherry. This approach does not only result in the implementation of foreign genes but also gives the possibility to differentiate between cell types via fluorescence microscopy [57]. Independent of the kind of foreign gene sequence, it is possible with the help of specific primers to quantify this non-human gene sequence to quantify the cells in the co-culture approach. A pitfall of this approach is the necessary efficiency of nearly 100% for both primers, which can only be reached using qPCR with perfectly optimized primers.

Nevertheless, our newly developed PCR-based co-culture quantification method has limitations. For example, it cannot be used in spheroid cultures. In the spheroid interior, there is a necrotic core that contains DNA and would lead to false-positive results [58]. Dead cells therefore should always be removed prior to DNA isolation.

Despite the limitations mentioned above, this quantification method can find, as summarized in Table 4, broad application in recently published co-culture approaches of different organ systems.

## 5. Conclusions

As our data demonstrate, we developed a PCR-based 3D co-culture quantification method that allows the quantification of two different cell types (of human and mouse origin) accurately. In comparison to conventional quantification methods (such as resazurin conversion), this method not only enables the separate quantification of the individual cell types but also improves the comparability of 2D to 3D culture results by excluding culture-related differences in metabolism.

## Figures and Tables

**Figure 1 mps-03-00001-f001:**
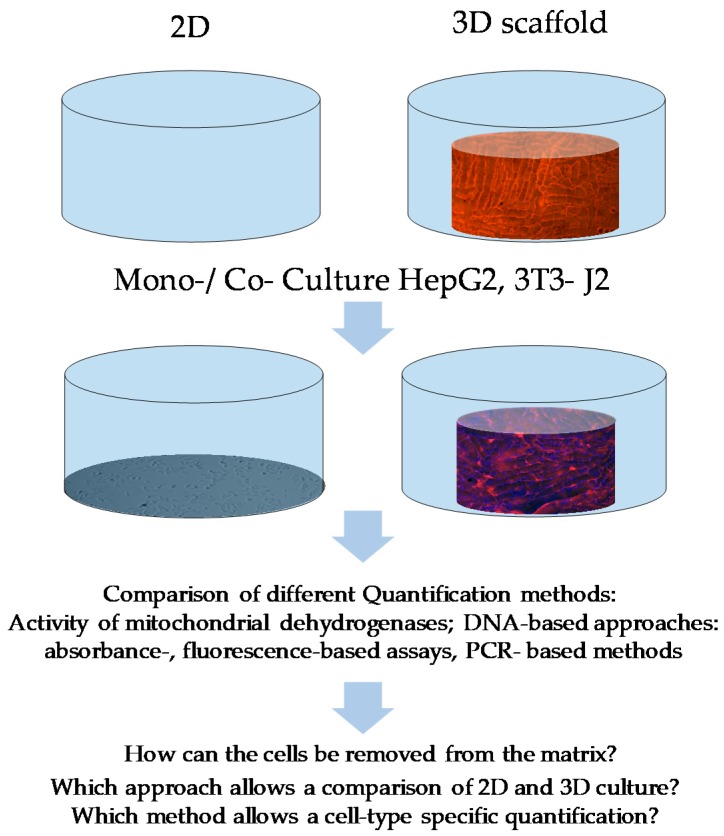
Comparison of different methods for the quantification of HepG2 and/or 3T3-J2 in mono- or co-culture. Three-dimensional culture environment Optimaix-3D scaffolds are used. The Optimaix-3D scaffolds (height 1.5 mm, Ø 5 mm) have a mean pore size of 88.9 ± 21.1 µm and a porosity of 96.3 ± 0.3% as described before [4]. Representative pictures of HepG2/3T3-J2 cells plated on scaffolds are shown in Appendix A.

**Figure 2 mps-03-00001-f002:**
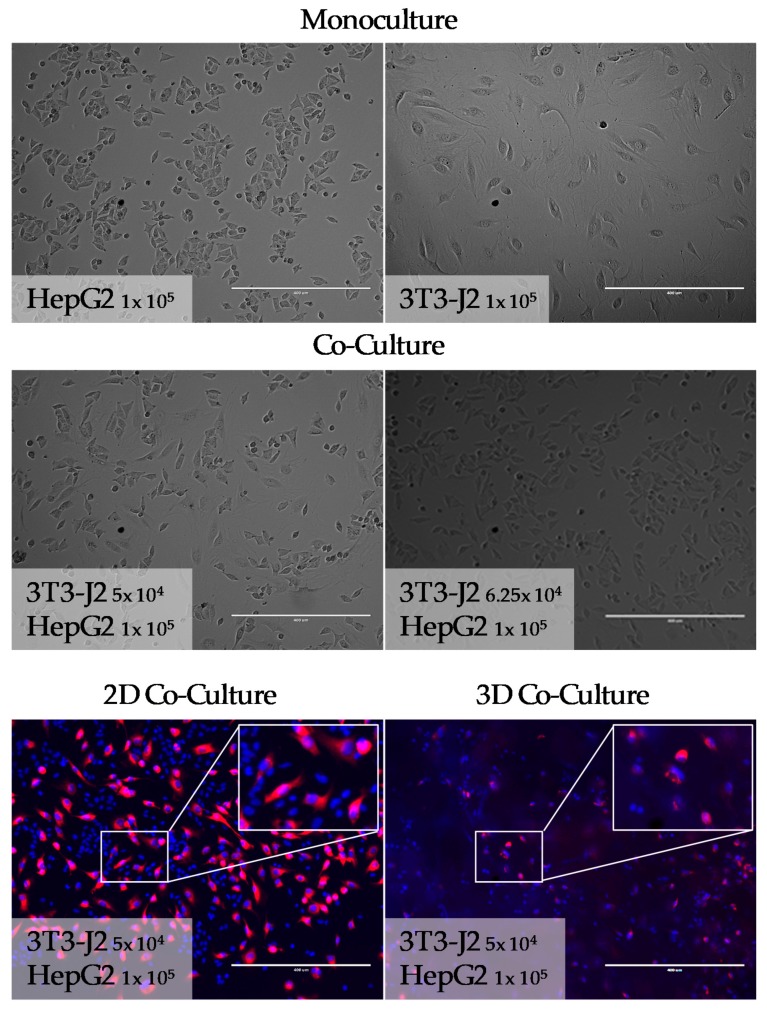
Light microscopy images of HepG2 cells and 3T3-J2 cells in mono- and co-cultures. HepG2 cells and 3T3-J2 cells were plated out using the cell numbers indicated in the pictures. Images were taken 18 h after plating the cells. For the fluorescence images, a carbocyanine dye was used to stain the cell membranes of the 3T3-J2 cells (red). All cell nuclei were stained with Hoechst 33342 (blue) and 100-fold magnification was used for all images. Scale bar is 400 µm.

**Figure 3 mps-03-00001-f003:**
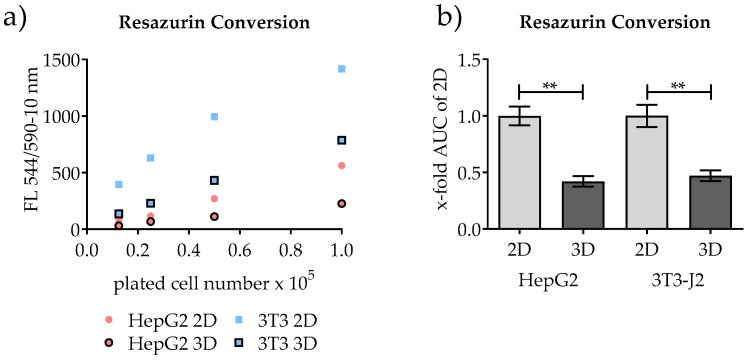
Shown here are 2D culture and 3D culture quantification by measurement of resazurin conversion. (**a**) Resazurin conversion of mono-cultures of HepG2 cells and 3T3-J2 cells in correlation to the plated number of cells. (**b**) Calculated area under the curve (AUC) of the resazurin conversion in HepG2 cells and 3T3-J2 cells (3D culture compared to the conventional 2D culture). *n* = 3, *n* = 4 (three independent runs, four technical replicates for each run); mean ± SEM; *** p* < 0.01.

**Figure 4 mps-03-00001-f004:**
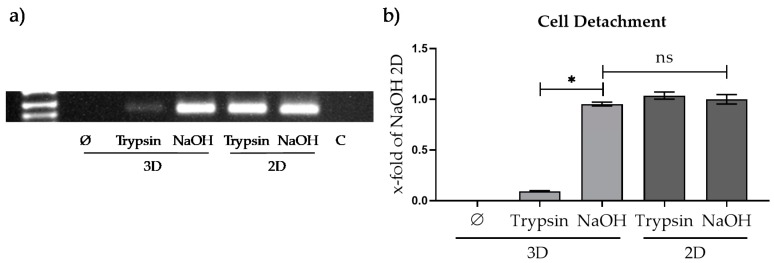
Cell detachment experiment, HepG2 cells were seeded onto Optimaix-3D scaffolds and in 2D culture plates. On the next day by PCR, it was tested whether the cells can be detached from the scaffold by centrifugation only or via incubation with trypsin. Additionally, cells seeded on scaffolds were lysed using pre-heated NaOH. As controls, 2D cultured cells treated with trypsin or NaOH were used. The number of cells (DNA) was determined by performing a conventional PCR targeting *hUGT1A6* gene (using the respective primer). (**a**) Representative gel picture of the *hUGT1A6* PCR using pooled samples of *n *= 3, *n *= 3. (**b**) *n* = 3: three scaffolds from each run were pooled and measured together within two technical replicates; mean ± SEM; * *p* < 0.05.

**Figure 5 mps-03-00001-f005:**
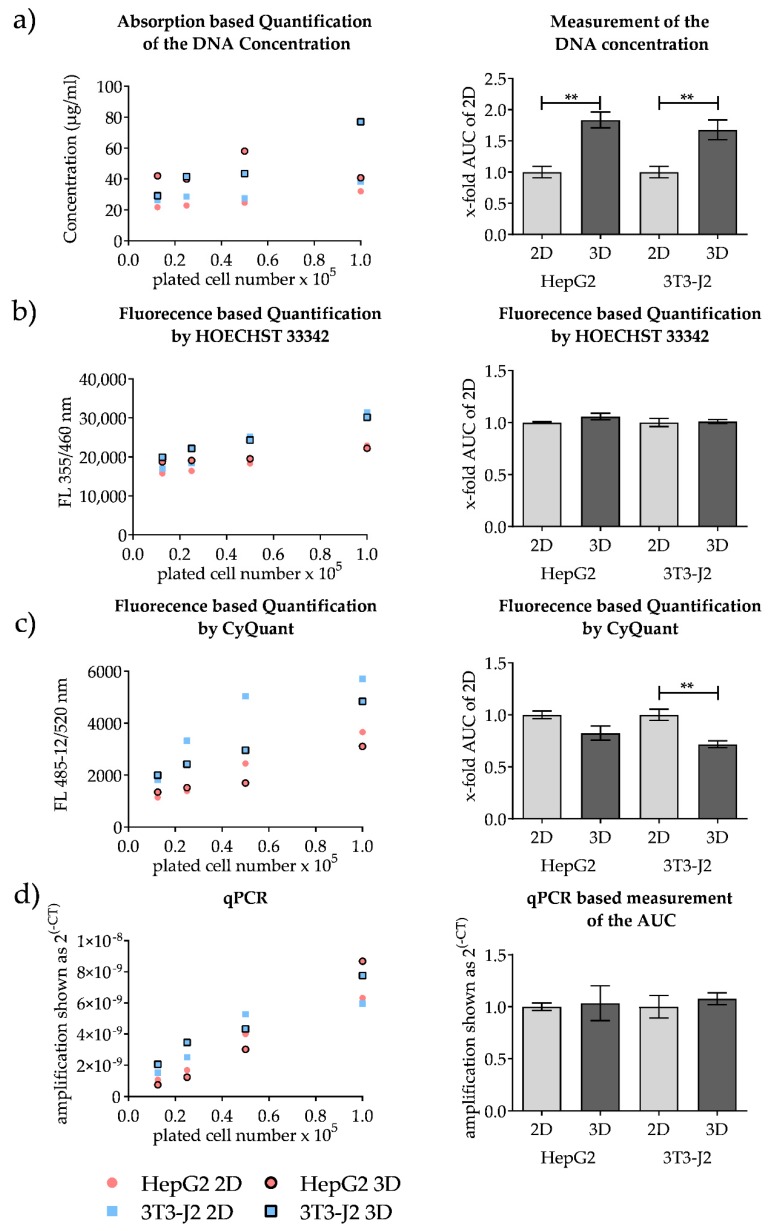
Comparison of different DNA-based approaches for 2D and 3D scaffold cell culture quantification. HepG2 cells and 3T3-J2 cells were plated in 2D and 3D as mono-cultures using different cell numbers. After an incubation time of 18 h, cells were lysed and the DNA was isolated. Four different quantification techniques were tested. The measured values of each quantification technique and the calculated area under the curve (AUC) are shown. (**a**) Absorption-based DNA quantification, (**b**,**c**) fluorescence-based DNA measurement using Hoechst 33342 and CyQuant, and (**d**) quantification of absolute cell numbers by qPCR with cell-type specific primers. *n* = 3, *n* = 2 (three independent runs with four replicates, pooled to two technical replicates prior DNA isolation); mean ± SEM; ** *p* ≤ 0.01.

**Figure 6 mps-03-00001-f006:**
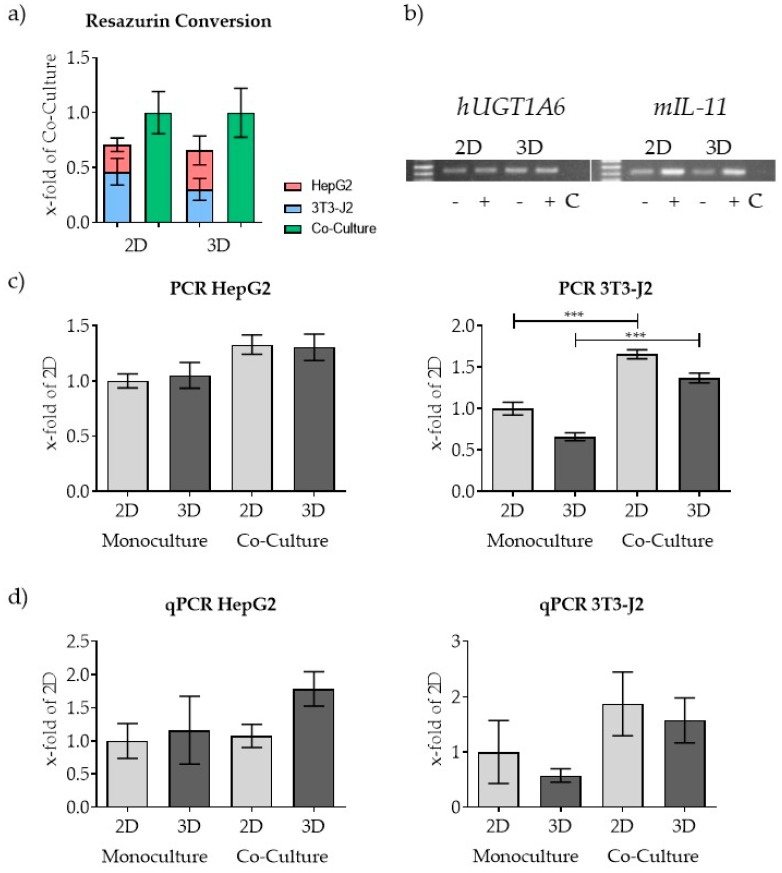
Application of the newly developed PCR-based quantification method on a co-culture consisting of 5 × 10^4^ HepG2 and 5 × 10^4^ 3T3-J2 cells. Both cell lines were plated in co-cultures and in mono-cultures. (**a**) Measurement of the resazurin conversion of the mono- and co-cultures after 1 h incubation. (**b**) Representative figure of the pooled samples analyzed by conventional PCR (co-cultures (+) and the mono-cultures (–)) also showing the negative signal of the control (**c**), which consists of DNA of the 2D mono-culture from the other respective cell line. Images of gels used for analysis can be found in Appendix A. The amount of DNA for each cell type was measured by conventional PCR (**b**,**c**) and by qPCR (**d**) using cell-type-specific primers. *n* = 4, *n* = 2; mean ± SEM; *** *p* < 0.001.

**Figure 7 mps-03-00001-f007:**
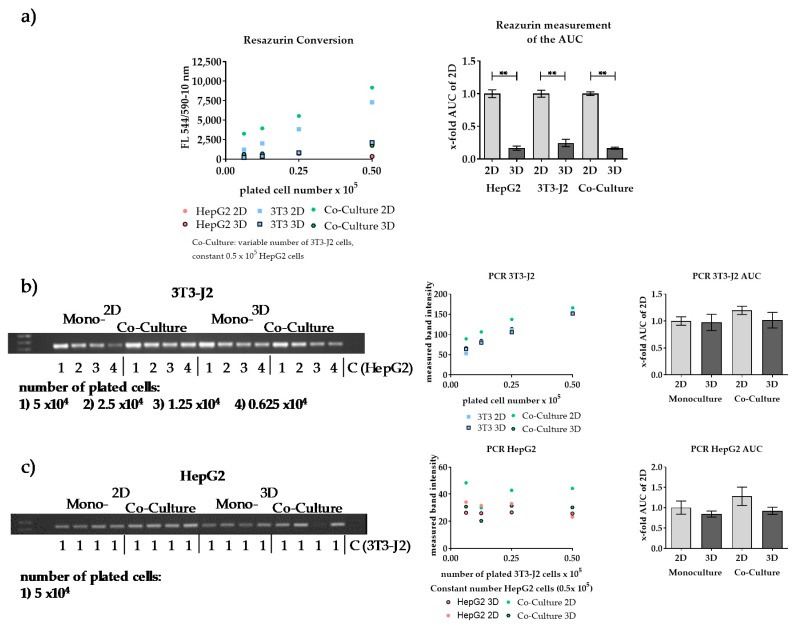
Transfer of the newly developed cell-type-specific quantification method to a self-made cryogel scaffold. Both cell lines were plated in co-cultures and in mono-cultures. Therefore, a constant number of HepG2 cells (5 × 10^4^ cells) and a variable number of 3T3-J2 cells (5 × 10^4^–6.25 × 10^3^ cells) were plated on scaffolds as well as in 2D. (**a**) Resazurin conversion of the mono- and co-cultures was measured. (**b**,**c**) The DNA amount of each cell type was measured by conventional PCR using species-specific primers. (**b**) 3T3-J2 cells with *mIL-11* primers. (**c**) HepG2 cells with *hUGT1A6* primers. *n* = 3, *n* = 2; mean ± SEM; ** *p* < 0.01.

**Table 1 mps-03-00001-t001:** Comparison of different cell quantification techniques.

	Mitochondrial Activity (Resazurin, MTT, or XTT)	ATP Measurement	LDH Measurement	DNA Staining (CyQuant)	Protein Staining (SRB)	Protein Quantification (Lowry)
**Assay principle**	Measurement of mitochondrial dehydrogenase activity	Measurement of total ATP levels	Measurement of released LDH	Staining of total DNA	Staining of total protein	Measurement of soluble protein
**Advantages**	Wide range of applications, distinction of dead and living cells is possible	Very sensitive assay, distinction of dead and living cells possible, not affected by stress level of the cells	Very sensitive assay, not affected by stress level of the cells	Very sensitive assay, not affected by stress level of the cells	Favorable and stable assay for quantification of adherent cells, not affected by stress level of the cells	Used for adherent and suspension cells
**Disadvantages**	Affected by stress level of the cells	Lysis of cells necessary	Lysis of cells necessary for normalization, assay is susceptible to interference (e.g., by FCS)	Lysis of cells necessary, assay is susceptible to interference (e.g., by phenol red of the medium)	Cannot be used for quantification in 3D culture	Lysis of cells necessary, assay is susceptible to interference (e.g., by FCS or scaffold ingredients)
	**No distinction between the different cell types in co-culture possible**					
**References**	[4,6,16,17]	[8]	[18]	[7]	[12,13,19]	[11]

**Table 2 mps-03-00001-t002:** Primers used in species-specific DNA amplification.

Gene	*mIL-11*	*hUGT1A6*
Forward-Sequence 5′-3′	TGCTGACAAGGCTTCGAGTAG	TGGTGCCTGAAGTTAATTTGCT
Reverse-Sequence 5′-3′	ACATCAAGAGCTGTAAACGGC	GCTCTGGCAGTTGATGAAGTA
Amplicon in bp	156	209
Annealing Temperature in °C	62	62
Cycle Number	30	30
Reference Sequence	NC_000073.6	NC_000002.12

**Table 3 mps-03-00001-t003:** Limits of Detection (LOD), Limits of Quantitation (LOQ), and sensitivity of the tested DNA-based quantification methods.

Method		Number of Cells		
	Cell Line	LOD	LOQ	Sensitivity (%)
Absorption-based quantification	HepG2	2183	7277	95
	3T3-J2	2557	8523	98
Fluorescence-based quantification (HOECHST 33342)	HepG2	5291	17,635	88
	3T3-J2	3400	11,334	98
Fluorescence-based quantification (CyQuant)	HepG2	1506	5018	104
	3T3-J2	471	1571	101
qPCR-based quantification	HepG2	1742	5808	99
	3T3-J2	1447	4824	99

The standard curves shown in Appendix A were used to calculate the analytical figures of merit. The Limit of Detection (LOD) and Limit of Quantitation (LOQ) were calculated as three-/ten-times of the standard error of the y-intercept. The slope of the standard curve showed the sensitivity of the respective method.

**Table 4 mps-03-00001-t004:** Possible applications for the PCR-based co-culture quantification method. Shown is a selection of publications in which cells from different species were used in a co-culture approach. For a combination other than human/mouse cells, another species-specific primer needs to be used.

Organ System	Cell Type I	Cell Type II	Ref
Liver (HEPATOPAC^®^)	Primary human hepatocytes	m3T3-J2	[25]
Liver	HepG2	m3T3-J2	[26]
Liver	primary rat hepatocytes	m3T3-J2	[27]
Liver	human cord blood stem cells	hepatic alpha mouse liver 12 cells	[59]
Nervous system	human oral mucosal stem cells	mouse neural stem cells	[60]
Bone	SaOS2	RAW 264.7 cells	[61]
Cartilage	bovine primary chondrocytes	Three different cell lines from mouse/human	[62]

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
