# Peer review of "Cell-Type-Specific Quantification of a Scaffold-Based 3D Liver Co-Culture"

_mps, 2019, doi:10.3390/mps3010001_

Round 1

Reviewer 1 Report

The paper is difficult to read and to understand. Professional English editing is strongly recommended.

It is not clear what the authors' aim is by mixing two cell lines from different species.

Author Response

We'd like to thank the reviewer for carefully reading the manuscript and addressing relevant questions. The detailed response to the reviewers questions is attached.

Reviewer 2 Report

I find this work to be both timely and compelling, as many groups are making the shift from monolayer to 3D cultures. The authors cover a large number of techniques and show that DNA content is a means of quantifying cells without fear of changes in the cellular environment or stress.

To be more informative, and help others make the appropriate decision when thinking of the best quantification method the authors should:

1) Discussing the limitations of the approach when generating co-cultures of human cells, making it not possible to use PCR-based techniques. There are some really nice examples of DNA barcoding in the literature, where groups have used this to track individual cells or cell lineages in xenograft models, to track cellular invasion in 3D cultures containing multiple cell types, and when evaluating the evolution of tissues.

--> https://pubs.acs.org/doi/10.1021/acs.analchem.5b02362

--> https://journals.plos.org/plosone/article?id=10.1371/journal.pone.0067316

--> https://www.nature.com/articles/ncomms6871.pdf

--> https://www.sciencedirect.com/science/article/pii/S1934590913005602?via%3Dihub

2) The authors do a great job of explaining the conceptual limitations of each measurement, however analytical figures of merit are not reported. For example, what are the limit of detection (LoD) and limit of quantitation (LoQ) for each method?  Also, what is the analytical sensitivity of each method and overall precision of the measurements for a given cell number in the co-culture format?

Without these figures of merit (and others such as linear range and overall sensitivity of the measurement), it does make it hard to determine when each method will be experimentally useful to others who want to use this method. 

3) The authors state that is it difficult to ensure all the cells are removed from a 3D scaffold. Some complementary data showing that cells are retained in their setups would be very helpful to determine what percentage of the cells remain and if these cells could skew results when not using a lyse/analyze PCR-based method.

4) In Figures 2a, 3a, 3b and 3c there are no error bars associated with the fluorescence ration. These should be included as well as statements about the numbers of samples run.

5) Figure 4b seems to contain no information. A more thorough analysis of the PCR data with images of each gel (with the band of interest and a loading control) should be placed in the SI.

6) The 3D scaffold should be better described (thickness, overall porosity) and images of cells inside the scaffold provided. One could argue this paper is not about the scaffold chosen but more about 2D vs. 3D.  I agree with taht argument but also think it is hard to assess data obtained from a 3D cell culture system that does not contain representative images of the setup.

Author Response

(The authors gave the same response as above.)

Reviewer 3 Report

The current paper (mps-537594) deals with technical and experimental investigations aiming at separately and accurately quantifying different cell types in 3D culture systems. Based on data recovered, the authors claim the development of a PCR-based method that allows a discriminated quantification of different cell populations 

The ms is ok and easily readable. The data are presented in a clear way. Only the legends of the supplementary figures should be more detailed 

Few comments are here detailed:

The paragraph 3.1 should be presented as an introduction one for the results and not a s a whole paragraph with no data described or presented What is the concentration of glucose in the DMEM used? What is the supplier of FCS? What is the concentration of trypsin used? Page 4, line 132: change "hot" by "heated" Supplementary figure 1: the authors should add the magnification of the images used Page 8, line 253: the sentence is not clear to me and should be rewritten Page 16, line 442: omit the reference 54

Author Response

Dear Editor, dear Reviewers,

We thank you, for the opportunity to revise our manuscript.

We appreciated your comments. We have addressed all issues regarding your recommendations and edited the manuscript accordingly. The detailed answers to your questions are summarized below. All changes performed in the manuscript are highlighted.

We would like to resubmit our revised manuscript to the journal Methods and Protocols and hope that the revised manuscript meets now the high standards of this journal.

All authors have approved the final submitted manuscript and no conflict of interest exists between the authors.

Yours sincerely,

Marc Ruoß

Reviewer 4 Report

The manuscript by Ruoß and colleagues refers to a problem, which is becoming increasingly important in respect to optimizing cellular performance in culture systems. This is relevant in pharmacological testings for animal replacement approaches and includes both the maintenance of cellular functions, esp. when using primary cells and the conditioning of cells prior to cellular therapies, esp. when using stem cell-derived cells for therapy stratification. The conclusion of the authors favours a PCR-based approach using species-specific primers for (semi)quantitative RNA analysis. This is, of course, only feasible when cells from different species were used, a limitation, which is also discussed by the authors. The only thing missing might be to discuss the relation of functional assays relying on protein(concentrations), which must not necessarily correlate to RNA levels.

Overall, though the experimental design of the study is clear cut, the presentation of results is sometimes blurry, because mostly Figures are only referred to as a whole instead of referring to parts of the figures where they are mentioned in the text.

Specific comments:

For the co-culture experiments, please indicate cell numbers of the individual cell types (line 92 and 99).

Remove highlights from all text passages.

Please check spelling in descriptions of Figure 1 (e.g. absorbance-, fluorescence-based, PCR-based etc.).

Legend to Supp Fig. 2: species-specify?

Section 3.1 does not show any data. What is the intention of this section?

Symbols in Figure A1 are non-readable. In addition, since it is not mentioned, e.g., line 253, to which part of the Figure A1 the text refers to, this figure is hard to understand. Maybe, sub-numberings like A1a, A1b etc. should be introduced. Why the designation of symbols and lines is repeated, one with and one without dots?

Figure 4 a: What is shown? a) PCR bands or total DNA on an agarose DNA gel? Line 287: by? Line 291: and?

Lines 302-303: this statement is not consistently true and should be modified accordingly.

Line 298-308: Please refer more specifically to Fig. 5 A-C

Please label Y-axis in Fig. 5D.

Table 3: an? DNA-based? Quantification? Which figures have been used for the slope and how it was calculated?

Lines 328-334: What is the conclusion out of this experiment?

Please re-phrase sentence lines 341-343. PCR is not feasible to measure resazurin conversion.

If the gel shown in Fig. 6b is one out of those shown in Suppl. Fig. 4, it should be shown there as well. At least the mIL-11 run is not presented in Suppl. Fig. 4 (different marker lane).

Line 366: disturbs?

There is some confusion with Fig. 7c. Is it correctly that different cell numbers were used for plating as shown in the PCR quantification on the right side? If yes, the gel run does not fit. In addition, the legend of the X-axis says 3T3-J2 cells; aren’t we talking about HepG2 cells?

Line 460: Table 1?

Please refer to Table 4 somewhere in the main text.

Author Response

(The authors gave the same response as above.)

Round 2

Reviewer 1 Report

Although the authors have made some improvements in the paper, the study itself has yet several flaws and the experiments lack validity.

First, the authors are looking at mRNA levels of mouse-IL-11 (NM_001290423.1) and human-UGT1A6 (NM_001072.3). The authors are make confusion about RNA and DNA. Moreover, these mRNA expression levels could change during culture. More stable internal controls should also be evaluated. 

English writing has improved but still very poor. There are numerous typographic errors. Additionally, although grammatically correct, there are many sentences that are not well elaborated, and does not reach an appropriate level, as a scientific paper should have. 

Author Response

(The authors gave the same response as above.)

Reviewer 4 Report

The manuscript improved considerably by the revision. There are only some points, which this reviewer thinks, need attention.

Legend to Supp Fig. 2: species-specify =-specificity?

The addition of Figure A1 at the end of the manuscript is highly confusing. The reader does not expect a figure at the end of the manuscript. If the authors think that the figure is needed (because it does not tell anything additional than shown in Figg 3 and 5), it is recommended to shift it into the supplements.

At line 251, the reader is asking: What does the A1 stand for? Why not just say: Figure 3a?

Line 295: ibid. What means Figure A1? And further (b-e), Figure 5 only shows a-d.

Author Response

Dear Editor, dear Reviewers,

We like to thank the editor and reviewers for giving us the opportunity to re-revise our manuscript. We appreciated your comments and have addressed all issues regarding your recommendations and edited the manuscript accordingly. The detailed answers to the reviewer comments are summarized below. All changes performed in the manuscript are highlighted. We would like to resubmit our revised manuscript to Journal Methods and Protocols and hope that the revised manuscript meets now the high standards of this journal.

All authors have approved the final submitted manuscript and no conflict of interest exists between the authors

Yours sincerely,

Prof. Dr. Andreas K. Nüssler
